# Model Systems to Study the Mechanism of Vascular Aging

**DOI:** 10.3390/ijms242015379

**Published:** 2023-10-19

**Authors:** Janette van der Linden, Lianne Trap, Caroline V. Scherer, Anton J. M. Roks, A. H. Jan Danser, Ingrid van der Pluijm, Caroline Cheng

**Affiliations:** 1Division of Vascular Medicine and Pharmacology, Department of Internal Medicine, Erasmus MC, 3015 GD Rotterdam, The Netherlands; 2Department of Molecular Genetics, Cancer Genomics Center Netherlands, Erasmus MC, 3015 GD Rotterdam, The Netherlands; 3Department of Pulmonary Medicine, Erasmus MC, 3015 GD Rotterdam, The Netherlands; 4Department of Internal Medicine, Erasmus MC, 3015 GD Rotterdam, The Netherlands; 5Department of Vascular Surgery, Cardiovascular Institute, Erasmus MC, 3015 GD Rotterdam, The Netherlands; 6Division of Experimental Cardiology, Department of Cardiology, Erasmus MC, 3015 GD Rotterdam, The Netherlands; 7Department of Nephrology and Hypertension, Division of Internal Medicine and Dermatology, University Medical Center Utrecht, 3584 CX Utrecht, The Netherlands

**Keywords:** vascular aging, genomic instability, mouse models, vessels-on-a-chip

## Abstract

Cardiovascular diseases are the leading cause of death globally. Within cardiovascular aging, arterial aging holds significant importance, as it involves structural and functional alterations in arteries that contribute substantially to the overall decline in cardiovascular health during the aging process. As arteries age, their ability to respond to stress and injury diminishes, while their luminal diameter increases. Moreover, they experience intimal and medial thickening, endothelial dysfunction, loss of vascular smooth muscle cells, cellular senescence, extracellular matrix remodeling, and deposition of collagen and calcium. This aging process also leads to overall arterial stiffening and cellular remodeling. The process of genomic instability plays a vital role in accelerating vascular aging. Progeria syndromes, rare genetic disorders causing premature aging, exemplify the impact of genomic instability. Throughout life, our DNA faces constant challenges from environmental radiation, chemicals, and endogenous metabolic products, leading to DNA damage and genome instability as we age. The accumulation of unrepaired damages over time manifests as an aging phenotype. To study vascular aging, various models are available, ranging from in vivo mouse studies to cell culture options, and there are also microfluidic in vitro model systems known as vessels-on-a-chip. Together, these models offer valuable insights into the aging process of blood vessels.

## 1. Introduction

Cardiovascular diseases (CVDs) are the leading cause of death worldwide, especially in the elderly [1]. In 2019, CVD affected 523 million people and killed 18.6 million people globally. Modifiable risk factors for developing CVD include high systolic blood pressure, high fasting plasma glucose, high low-density lipoprotein cholesterol, and smoking [2]. Blood pressure and pulse wave velocity measurements, which assess arterial stiffness, are used to predict cardiovascular events and mortality, and increase in prevalence with age [3,4,5]. Arterial aging is a fundamental aspect of cardiovascular aging, as the structural and functional changes in arteries significantly contribute to the overall decline in cardiovascular health associated with the aging process. Unlike atherosclerosis, arteriosclerosis develops spontaneously during aging independently from cardiovascular risk factors other than age, although it is strongly accelerated by hypertension and hyperglycemia [6]. Arterial aging results in structural and functional changes in the arterial wall (Figure 1). Which involves chronic, low-grade inflammation, extracellular matrix (ECM) remodeling including elastin degradation, medial thickening, endothelial dysfunction, phenotypical switching of vascular smooth muscle cells (VSMCs), calcification, and fibrosis [7]. Various molecular and cellular mechanisms contribute to aging of the vasculature (Figure 1). Reactive oxygen species (ROS) and impaired bioavailability of nitric oxide (NO), together with reduced antioxidant defense systems, cause oxidative stress, leading to endothelial dysfunction and inflammation [8]. Vascular inflammation, mitochondrial dysfunction, cellular senescence, apoptosis, deregulated nutrient sensing, and genomic instability also play a major role in vascular aging [8]. Accordingly, most of these factors are considered to be hallmarks of aging [9,10].

In this review, we describe different parameters of vascular aging and the way that different model systems are used to investigate the underlying molecular mechanism of vascular aging. We will focus on the different types of models to study vascular aging, ranging from in vivo mouse studies to cell culture options and microfluidic in vitro model systems, also known as vessels-on-a-chip.

## 2. Structural, Functional, and Pathological Changes in the Aged Vascular System

Aging of arteries is characterized by structural changes (Figure 1); these include increase in luminal diameter, intimal and medial thickening, loss of VSMCs, senescence, ECM remodeling, collagen and calcium deposition, and cellular remodeling [11]. These structural alterations result in functional decline of the vascular system. As a result of these structural changes, older adults display physiological changes, including endothelial dysfunction, arterial stiffening, and high blood pressure [12]. Pathological changes can also occur during aging; these include arteriosclerosis, aneurysms, and heart failure.

### Cellular Changes in the Aged Vascular System

Mitochondria, being the primary source of ROS production, play a significant role in promoting oxidative stress and low-grade inflammation during vascular aging. With age, there is an increase in ROS production, decreased antioxidant capacity, and impaired cellular redox signaling. Excessive ROS may lead to endothelial dysfunction, inflammation, and structural damage to the vascular wall [8]. Chronic low-grade inflammation is a characteristic feature of aging. In vascular aging, upregulation of inflammatory markers, increased infiltration of immune cells into the vessel wall, and activation of pro-inflammatory signaling pathways are observed. Arterial stiffness is a hallmark of vascular aging and is partially attributed to changes in the elastic fibers and collagen content of the arterial walls. With aging, there is an accumulation of collagen and other extracellular matrix proteins in the arterial wall, leading to reduced arterial compliance and increased arterial stiffness. This contributes to impaired arterial distensibility and elevated systolic blood pressure. Arterial distensibility is also influenced by endothelial dysfunction, the inability of arteries to maintain proper vascular tone. The processes of vasodilation and its counterpart vasoconstriction are essential to regulate blood flow and the supply of oxygen and nutrients throughout the body. NO-cyclic guanosine monophosphate (cGMP) signaling is crucial for proper functioning of the vasculature. Endothelial cells produce NO, which then diffuses to VSMCs and activates soluble guanylyl cyclase (sGC)-mediated production of cGMP, finally leading to vasodilation, e.g., relaxation and widening of blood vessels (Figure 2). The eNOS protein, together with cofactor tetrahydrobiopterin (BH4), synthesizes NO through the oxidization of nicotinamide adenine dinucleotide phosphate (NADPH) and the conversion of L-Arginine. ROS, which are abundant in the case of oxidative stress, are capable of scavenging NO, thereby decreasing NO bioavailability. Endothelial dysfunction occurs when maintenance of vascular tone is impaired (Figure 2). This is caused by a shortage of endothelium-derived relaxing factors, like NO, or by a decrease in VSMC responsiveness to NO. A decrease in NO bioavailability likely results from uncoupling of endothelial nitric oxide synthase (eNOS), which results in decreased production of NO and increased generation of ROS. Furthermore, it increases NADPH oxidase (NOX) activity, which results in increased ROS production, or increased production of ROS by mitochondria. Phosphodiesterase (PDEs) catalyze hydrolysis of cGMP and cAMP, leading to decreased responsiveness to NO (Figure 2).

Other vasodilator signaling pathways of the endothelium are endothelium-derived hyperpolarization (EDHF) and prostaglandins (PG). The faith of EDHF in aging is unknown. For PG, it is known that in aging there is a switch from dilatory PG, such as prostacyclin, toward vasoconstrictive PG [13]. However, PG-mediated responses are often not observed in mouse models, and are, therefore, difficult to study in genetic aging models.

Furthermore, aging contributes to phenotypical switching of VSMCs. Several VSMC phenotypes have been described; contractile, synthetic, osteogenic, senescent, macrophage-like, mesenchymal stem cell-like, fibroblast-like and adipose-like (Figure 3) [14,15,16,17]. The contractile phenotype expresses contractile markers such as SMA, SM22, and MYH11 [18]. These cells can dedifferentiate towards a more proliferative and migrating phenotype, the synthetic VSMCs which secrete high levels of ECM, and are positive for vimentin, EREG, and MMP9 [18,19]. Osteogenic VSMCs are characterized by increased levels of RunX2, OPN, and ALP [20]. These cells contribute to vascular calcification. Macrophage-like VSMCs are positive for LGALS3, CD68, and VCAM1. These cells are capable of initiating an immune response and are associated with atherosclerosis [21]. The fibroblast-like VSMCs are associated with fibrosis of the vascular wall. They express LUM, BGN, and DCN [16]. Senescence is a hallmark of aging; senescent VSMCs express increased P16, P21, and senescence-associated secretory phenotype (SASP) markers [14]. Mesenchymal stem cell-like VSMC express SCA-, CD34, and CD44. The role of mesenchymal stem cell-like VSMC is not yet known [14]. The adipocyte-VSMCs are only described in single cell RNA-seq [16]. They express markers such as UCP1, DIO2, and PPARGC1. More research is needed to unravel the function of this phenotype.

The phenotypical switching of VSMCs occurs upon various stimuli. Endothelial denudation or atherogenic lesions have been well-studied in this respect. The impact of endothelial aging is not as clearly studied directly in aging [22,23]. However, decreased NO signaling, a hallmark of endothelial aging, has been studied. ECs synthesize and release NO, which induces relaxation in VSMCs while also preserving the quiescent state of VSMCs via inhibition of proliferation by cGMP, the main messenger of NO [24,25,26]. Furthermore, NO–cGMP signaling decreases VSMC-mediated fibrosis and prevents medial hypertrophy. Hence, improved NO signaling contributes to vasodilation and maintenance of vascular compliance, thus sustaining proper tonus and flow regulation via VSMC [27]. Loss of NO is likely to lead to switching to the pro-fibrotic, osteogenic, and synthetic VSMC phenotypes.

## 3. Vascular Aging in Human Progeria Syndromes

Genomic instability contributes to accelerated vascular aging in humans. Progeria syndromes are a group of rare genetic disorders that result in premature aging, with a range of health problems that are typically associated with advanced age. Cardiovascular disease is a major cause of morbidity and mortality in patients with progeria syndromes, and several types of progeria syndromes have been identified that manifest juvenile cardiovascular diseases.

Hutchinson–Gilford Progeria Syndrome (HGPS) is the most well-known progeria syndrome and is caused, in most patients, by a mutation in the LMNA gene, which encodes for lamin A/C. Lamins play a crucial role in providing structural support to the nucleus, and their expression levels are directly associated with nuclear stiffness, tissue rigidity, and plasticity. Furthermore, lamin A contributes to genomic instability. Lamin A participates in maintaining telomere homeostasis and plays a role in DNA repair processes by silencing DNA repair proteins like BRCA1 and RAD51 [28,29]. HGPS is characterized by premature aging, including accelerated cardiovascular disease, which is the leading cause of death in these patients. HGPS patients typically have loss of VSMCs, arterial stiffness, and calcification of the arterial walls. Interestingly, they do not develop fully formed plaques, but the initial phase of atherosclerosis. These pathological alterations can lead to myocardial infarction, stroke, and premature death, mainly caused by arteriosclerosis [30,31]. The average age of death for children with HGPS is 13 years [32].

Werner Syndrome (WS) is another progeria syndrome that is associated with increased risk of cardiovascular disease. WS is caused by a mutation in the WRN gene, which encodes for Werner syndrome ATP-dependent helicase. WRN is involved in DNA repair by unwinding 5′ flaps and interacting with base-excision repair proteins [33]. WS patients often develop atherosclerosis, which increases risk of myocardial infarction and stroke. Furthermore, these patients show calcification of the aortic valve. WS patients also have a high incidence of hypertension and heart failure, which further increases their risk of cardiovascular disease [34].

## 4. Vascular Aging Mouse Models

The fundamental processes behind aging have also been studied by using invertebrate animal models systems such as roundworms (*Caenorhabditis elegans*) or the fruit fly (*Drosophila melanogaster*) [35]. While these models are inexpensive and available in great quantity, mammalian model organisms better approximate the complexity of human aging. However, while some long-lived species such as primates are closest to humans, their generational cycles are so long that it is difficult to use them in research [36]. Mice, specifically genetically modified fast-aging mice, are therefore currently most often used for aging research.

Genetically engineered mouse models have been established that mimic human progeria syndromes [35]. In these progeroid mice, genes involved in genome maintenance are genetically manipulated, which leads to accelerated aging and a shortened lifespan. These accelerated aging models can be divided into different groups: genomic instability, telomere attrition, epigenetic alterations, loss of proteostasis, mitochondrial dysfunction, stem cell exhaustion, and altered intercellular communication. The majority of the gene mutations in these accelerated aging mouse models result in genomic instability. Furthermore, genomic instability is linked to all hallmarks of aging, highlighting the importance of DNA damage repair in aging [9].

Hence, although different types of mouse models are used to study vascular abnormalities, in this review, we will specifically focus on mouse models with genome instability that show accelerated vascular aging (Table 1).

### 4.1. Nucleotide Excision Repair (NER) Deficient Mouse Models

NER is a DNA repair pathway which eliminates a wide range of DNA lesions including single strand lesions and helix distorting lesions induced by exogenous sources such as ultraviolet radiation or by endogenous sources such as reactive metabolites [37,38]. Currently, *Ercc1^d/−^* and *Xpd^TTD^* mice are the only two NER deficient mouse models which have been described to show characteristics of vascular aging (Durik et al., 2012) [39]. DNA damage detection by NER occurs through two sub pathways: global genome NER (GG-NER), in which the whole genome is probed for DNA helix distortions, or transcription-coupled NER (TC-NER), which is activated by polymerase stalling due to lesions in the transcribed strand [39]. ERCC excision repair 1 (ERCC1) and Xeroderma Pigmentosum Group D (XPD) mediate the final part of the NER pathway, after which GG-NER and TC-NER converge. XPD is a helicase involved in verification of the DNA damage lesion and unwinding of the DNA around the lesion. The ERCC1 endonuclease forms a heterodimeric complex with Xeroderma Pigmentosum Group F endonuclease and excises the DNA strand 5′ of the lesion, in combination with Xeroderma Pigmentosum Group G (XPG), which excises 3′ of the lesion [38]. Ercc1 knockout mice lack expression of Ercc1 and live up to 38 days. On the other hand, *Ercc1^d/−^* mice–hemizygous for an Ercc1 knock-out allele and a delta -7 allele -resulting in ERCC1 protein with a seven amino acid truncation– live up to 6 months [40] and show accelerated age-related pathologies which chronologically follow changes in naturally aging wildtype mice [41]. These include neurodegeneration, muscular dystrophy, ataxia, brain atrophy, renal tubular degeneration, heart myocardial degeneration, peripheral nerve vacuolization, and several liver pathologies [42].

*Ercc1^d/−^* mice display increased aortic stiffness, blood pressure, aortic senescence, endothelial dysfunction, and decreased reactive hyperemia [39,41]—which is a measure for peripheral microvascular function used to predict cardiovascular morbidity and mortality.

Additionally, vascular aging has been studied in two tissue-specific Ercc1 knockout mouse models. Endothelial (EC)-specific Ercc1 knockout mice are hemizygous for a knock-out Ercc1 allele and a floxed Ercc1 allele and express Cre-recombinase under the Tie2 promoter [43]. Tie2 is expressed in ECs and in monocytes with a pro-angiogenic function. These EC-specific Ercc1 knockout mice exhibit decreased endothelium-derived NO, altered vasoconstriction, arterial remodeling, and increased arterial stiffness. Vascular smooth muscle cell (VSMC)-specific Ercc1 knockout mice are hemizygous for a knock-out Ercc1 allele and a floxed Ercc1 allele and express Cre-recombinase under the SM22α promoter [44]. SM22α is involved in SMC contraction. These mice show diminished NO-mediated vasodilation, decreased reactive hyperemia, and increased arterial stiffness.

Only three patients have been described with an ERCC1 mutation and all three died before the age of 3 [45,46,47]. No vascular aging phenotypes were described. However, it is unknown if these patients were screened for vascular problems at this very young age. As the *Ercc1^d/−^* mouse model shows characteristics of vascular aging, this would suggest that likewise patients with Ercc1 mutations may experience vascular aging problems with age.

### 4.2. LMNA

Multiple mouse models exist with mutations in the LMNA gene. The most commonly used is the *Lmna^G609G/G609G^* with a c.1827C>T;p.Gly609Gly mutation, which mimics a human c.1824C>T;p.Gly608Gly mutation in exon 11 [48]. The c.1827C>T;p.Gly609Gly mutation causes abnormal splicing, leading to the production of a shortened form of prelamin A, which serves as the precursor for lamin A/C. This truncated protein is commonly referred to as progerin. Progerin accumulates in the nuclear lumina, and thereby disrupts the nuclear architecture [49,50].

One study showed that the stiffness and inward remodeling observed in *Lmna^G609G/G609G^* arteries are primarily due to damage to VSMCs caused by ubiquitous progerin [50]. This leads to an increase in medial collagen deposition and subsequent changes to elastin structure [50]. The *Ldlr^−/−^Lmna^G609G/G609G^* mouse aortas showed similar results, including VSMC depletion, adventitial thickening, and changes to the elastin structure. They furthermore show atherosclerosis and structural abnormalities [51]. Coll-Bonfill et al. showed that progerin expression in VSMCs results in a phenotypical switch of the VSMCs (characterized by a decrease in contractile markers) towards a synthetic-like phenotype. This switch increases replication speed, resulting in replication stress, genomic instability and eventually VSMC depletion [52].

In the *Lmna^HG^* mouse model intron 10 and the last 150 nucleotide of intron 11 of the LMNA gene are deleted, resulting in the production of progerin [53]. Interestingly, both the *Lmna^HG/+^* (expressing progerin, lamin A/C) and the *Lmna^HG/HG^* mouse (expressing only progerin) do not show vascular aging [54]. This is in contrast with the c.1827C>T;p.Gly609Gly mutation, indicating a difference between the progerin protein resulting from the deletion of exon 10 and partially 11 and the c.1827C>T;p.Gly609Gly mutation, which mimics the most common human mutation, and gives a smaller deletion of only 50 amino acids within prelamin A, likely resulting in a differently folded protein.

Additionally, the process of vascular aging has been examined in mouse models that specifically target tissues of interest. To investigate the effect of VSMCs and ECs on vascular aging, del Campo et al. generated tissue specific mouse models, showing similar effects for lamin mutation in a VSMC specific mouse model (*Lmna^LCS/LCS^SM22αCre^+/tg^*^)^ compared to the full body mutant, but not in the EC-specific mouse model (*Lmna^LCS/LCS^Tie2Cre^+/tg^* mice) [50]. To further mimic the characteristics of HGPS, Hamczyk et al. crossed the *Apoe^−/−^* mouse with *Lmna^G609G/G609G^*. These mice, when fed a high fat diet, showed medial VSMC loss, lipid retention, adventitial fibrosis, and accelerated atherosclerosis [55]. Additionally, they showed a similar phenotype in the VSMC-specific mutant (*Apoe^−/−^Lmna^LCS/LCS^SM22αCre*), but not in the macrophage-specific mutant (*Apoe^−/−^Lmna^LCS/LCS^LysMCre*) [55].

Taken together, this shows that the *Lmna^G609G^* mutation promotes vascular aging in mice, while the *Lmna^HG^* mutant mice do not exhibit signs of vascular aging. The tissue-specific mouse models highlight the importance of Lmna in VSMCs, but no phenotypes of vascular aging were observed in the EC and macrophage KO mice.

This could indicate that VSMC dysfunction precedes and might trigger endothelial and macrophage dysfunction. Overall, defects in the nuclear lamina may generate genomic instability, and thereby trigger vascular aging. The observed vascular pathology closely mimics the majority of cardiovascular disease symptoms seen in HGPS patients [55].

### 4.3. Bub1b

BUB1B is involved in mitotic checkpoint regulation and the DNA damage response. Mutations in this gene have been associated with premature aging syndromes, such as mosaic variegated aneuploidy (MVA) syndrome. MVA is a rare genetic disorder characterized by abnormal chromosome numbers (aneuploidy) and mosaic patterns of chromosomal variation within different cells of an individual’s body. It is typically associated with severe developmental abnormalities, growth retardation, intellectual disability, and an increased risk of cancer. Studies in the BubR1 (Bub1b) mouse model have demonstrated that these animals display several features of accelerated aging, including in the vasculature.

Matsumoto et al. showed endothelial dysfunction, decreased levels of elastin, fibrosis, and thinning of both the arterial wall and the inner diameter, which was associated with a reduced number of VSMCs [56]. Another study showed increased senescence in the stromal vascular fraction, in particular, the fat progenitor cells [57]. Furthermore, *BubR1* mouse showed impaired angiogenesis [58]. Overall, these findings suggest a role of Bub1b, and thereby genomic instability, in vascular aging in mice, leading to endothelial dysfunction, ECM remodeling, and senescence.

### 4.4. Telomere Attrition

The Terc mouse is a commonly used model to study the effects of telomerase deficiency on aging and age-related diseases. Telomerase is an enzyme that maintains the length of telomeres, the protective caps at the end of chromosomes. Telomere shortening is a hallmark of cellular aging, and telomerase deficiency has been associated with accelerated aging and increased susceptibility to age-related diseases in humans. When telomeres become too short to function, it will trigger a DNA damage response in order to maintain genome stability.

It is important to mention the difference in telomere length between mice and humans, with mice having much longer telomeres. The generation and background of the Terc-deficient mice are crucial for vascular aging research, as phenotypes differ per generation [59]. The telomeres of the Terc-deficient mice shorten approximately 5 kb per generation. Most experiments are therefore performed in Terc-deficient mice after several generations, as older generations show more defects [60].

Several studies have investigated the effect of telomerase deficiency on vascular aging in Terc-deficient mice. One study found that *Terc*-deficient mice exhibit endothelial dysfunction and senescence, which is restored after antioxidant treatment, indicating a role for oxidative stress [61]. Another study showed that *Terc*-deficient mice experience hypertension as a result of increased endothelin-1 levels [62].

### 4.5. Epigenetic Alterations

SIRT6 is a protein that plays a crucial role in regulating cellular senescence and age-related cardiovascular diseases. SIRT6 is a histone deacetylase, and is involved in DNA repair by promoting the double strand break repair pathway non-homologous end joining (NHEJ), regulation of the expression of metabolic genes, and maintenance of genomic stability [63]. Studies have shown that SIRT6-deficient mice exhibit premature aging and accelerated vascular aging, characterized by endothelial dysfunction, inflammation, and increased oxidative stress. These mice also display increased susceptibility to age-related diseases, such as atherosclerosis and heart failure.

SIRT1, another deacetylase, protects against endothelial senescence and is involved in DNA damage recognition. SIRT1 decreases with aging, which is associated with endothelial dysfunction. Activators of SIRT1 such as Resveratrol are tested as intervention to prevent vascular aging [64].

SIRT6 is a key regulator of vasomotor function in conduit vessels, via tonic suppression of NAD(P)H oxidase expression and activation. It also contributes to endothelial dysfunction, growth arrest, and senescence. This phenotype can be rescued by overexpressing the transcription factor FOXM1, which is critical for cell cycle progression [65,66]. Another study showed that SIRT6 has a protective role in the vascular system, as downregulation results in vascular calcification [67]. Furthermore, *SIRT6^+/−^ApoE^−/−^* mice form atherosclerotic lesions and express the proinflammatory cytokine VCAM-1 [68]. These findings suggest that Sirt6 plays a critical role in promoting vascular aging as well as in protecting against age-related cardiovascular diseases.

### 4.6. Limitations of Mouse Models for Vascular Aging

None of the mouse models perfectly resembles human vascular aging. However, Table 1 shows an overview of the vascular phenotypes that are observed for each mouse model and are consistent with human vascular aging. Depending on the research question, the most suitable mouse model can be chosen. The *ERCC1*^d/−^ and *LMNA* mutant mice are the best characterized models and cell type specific mouse models are designed for both genes to further study the vascular aging.

Nevado et al. described premature vascular aging phenotypes in *Ldlr^−/−^Lmna^G609G/G609G^* and *Apoe^−/−^Lmna^G609G/G609G^* mouse models and observed a difference in atherosclerosis in normal aging and LMNA mice [51]. During normal aging, atheroma plaques start at one point and grow eccentrically, whereas in progeroid mice plaque formation occurs across almost the entire intimal surface, thereby making this mouse model less suitable to study atherosclerosis.

Matsumoto et al. describe the vascular phenotypes of *BubR1* mice, including endothelial dysfunction and VSMC loss (Table 1); however, only male mice were studied in detail [56].

A disadvantage of the *Terc*-deficient mice is that phenotypes differ per generation, making it more difficult to study vascular aging in this mouse model [59]. Furthermore, Terc-deficient mice display a shortened lifespan; however, they still live to over 1 year old, making them more time consuming compared to the other mouse models. On the other hand, *Sirt-6* deficient mice only live to 4 weeks of age, which makes intervention studies more difficult [69].

**Table 1 ijms-24-15379-t001:** Overview of vascular aging phenotypes in accelerated aging mouse models.

Mouse Model	Gene Targeted	Cell Type	Vascular Phenotypes	Reference
*ERCC1^d/−^*	*ERCC1*	Full body	Increased aortic stiffness, blood pressure, aortic senescence, endothelial dysfunction, and decreased reactive hyperemia	[39,40,41]
*ERCC1-Tie2Cre*	*ERCC1*	EC	Decreased endothelium-derived NO, altered vasoconstriction, arterial remodeling, and arterial stiffness	[43]
*ERCC1-SM22*	*ERCC1*	SMCs	Diminished NO-mediated vasodilation, decreased reactive hyperemia, and arterial stiffness	[44]
*Lmna^G609G/G609G^*	*LMNA*	Full body	Vascular stiffening, ECM remodeling	[48,50]
*Ldlr^−/−^Lmna^G609G/G609G^*	*LMNA*	Full body	VSMC depletion, adventitial thickening, and changes to the elastin structure, atherosclerosis	[51]
*Lmna^LCS/LCS^SM22αCre^+/tg^*	*LMNA*	SMCs	Vascular stiffening and ECM remodeling	[50]
*Lmna^LCS/LCS^Tie2Cre^+/tg^*	*LMNA*	EC	Unkown	[50]
*Apoe^−/−^Lmna^G609G/G609G^*	*LMNA*	Full body	VSMC depletion, adventitial thickening, and changes to the elastin structure	[55]
*Apoe^−/−^Lmna^LCS/LCS^SM22αCre*	*LMNA*	SMCs	VSMC depletion, adventitial thickening, and changes to the elastin structure	[55]
*Apoe^−/−^Lmna^LCS/LCS^LysMCre*	*LMNA*	Macrophage	Unkown	[55]
*Bub1b*	*Bub1b*	Full body	Endothelial dysfunction, decreased levels of elastin, fibrosis, thinning of both the arterial wall and the inner diameter, VSMC loss, and impaired angiogenesis	[56,58]
*Terc*	*Terc*	Full body	Endothelial dysfunction, senescence, and hypertension	[60,61,62]
*Sirt6*	*Sirt6*	Full body	Endothelial dysfunction and vascular calcification	[65,66,67]
*SIRT6^+/−^;ApoE^−/−^*	*Sirt6*	Full body	Atherosclerosis and vascular inflammation	[68]

## 5. The Role of Genomic Instability in Molecular and Cellular Mechanisms Underlying Vascular Aging

Our DNA is constantly challenged by environmental radiation and chemicals, as well as endogenous metabolic products causing DNA damage resulting in genome instability as we age. When these damages remain unrepaired and accumulate over time [9,10,70], they cause an aging phenotype. This damage to the DNA is further exacerbated in combination with oxidative stress which can trigger apoptosis or senescence, leading to the loss of ECs and VSMCs, thereby resulting in endothelial dysfunction, arterial stiffness, and arteriosclerosis (Figure 4) [71]. Contributors to vascular stiffening such as phenotype switching are also triggered by genomic instability as this changes gene expression in the ECM and vascular cells directly [72].

Genomic instability has been further associated with the dedifferentiation of VSMCs where they lose their contractile phenotype and shift towards a synthetic and osteogenic phenotype. The connecting factor here is the loss of expression of smooth muscle-specific genes, such as alpha-smooth muscle actin (SMA), muscle myosin heavy chain (Myh11), and calponin (Cnn1) (Figure 4) [73]. Instead, dedifferentiated VSMCs acquire markers of synthetic cells, like proliferative factors and extracellular matrix remodeling enzymes, including collagen and fibronectin. An increase in the expression of RUNX2 has also been linked to vascular aging [74,75]. RunX2 promotes calcification and stiffening of the arterial wall by inducing the expression of genes that regulate osteogenic differentiation (mineralization) and extracellular matrix (ECM) remodeling [15]. The ECM is the non-cellular component of the vessel wall that provides structural support to vascular cells and regulates vascular function. Changes in gene expression of ECM remodeling enzymes, such as matrix metalloproteinases (MMPs) and tissue inhibitors of metalloproteinases (TIMPs) affect the overall composition of the ECM causing a decrease in elastin content and an increase in collagen deposition, further contributing to vascular stiffness.

Dedifferentiation of VSMCs and ECM remodeling are interconnected processes. Dedifferentiated VSMCs secrete ECM remodeling enzymes, such as MMPs, that degrade the ECM and promote its remodeling. In turn, changes in ECM composition, such as collagen IV and laminin, affect VSMC phenotype and behavior by binding integrin receptors. Interaction between VSMCs, collagen IV, and laminin are important to keep the VSMCs in a quiescent state [76]. Furthermore, the interaction between VSMCs and ECM is affected by hemodynamic stress. Stiffening of the ECM leads to conformational changes in integrins, which plays an essential role in the response to pressure [77].

## 6. Models to Study Vascular Aging In Vitro

As mentioned earlier, mouse models exist with deficiencies in DNA repair mechanisms. Studying these models aids in comprehending the underlying mechanisms through which genomic instability leads to vascular aging. In order to gain more detailed insights at a cellular level and translate murine findings towards the human condition, it is necessary to utilize human in vitro models. In vitro models offer advantages such as easier scalability and the possibility to examine multiple parameters at the same time. Various approaches can be employed, such as utilizing human (patient derived) DNA repair-deficient vascular cells, as well as introducing endogenous DNA damage to these cells, in order to further investigate the mechanisms involved.

## 7. Cell Culture Models to Study Vascular Aging

Tissue-specific mouse models have significantly contributed to our understanding of vascular aging. These models are often designed to study specific cell types involved in the vasculature, such as ECs and VSMCs. ECs and VSMCs, in addition to pericytes and fibroblasts, are also commonly used in simple 2D cell culture assays, and, increasingly, also in complex 3D microfluidic tissue systems. Studying vascular aging in cell culture is an important approach to understand the molecular and cellular mechanisms underlying this complex process. Several parameters are commonly evaluated using (human) cell assays. Senescence is a hallmark of vascular aging and can be assessed through experiments such as the SA-β-gal activity assay or by detecting SASP markers [78]. Measuring the proliferation rate of cells can be useful to determine senescence and to investigate the phenotypical switching of VSMCs in vascular aging. Immunofluorescent staining and western blot are used to further evaluate the state of VSMCs, including the phenotype of the VSMCs by detecting, for example, SMA (contractile VSMC), RUNX2 (osteogenic VSMC), MAC2 (macrophage-like VSMC), and SCA1 (mesenchymal-like VSMC).

Structural changes in the ECM are also a hallmark of vascular aging and are studied in vitro by measuring the expression and activity of MMPs or other ECM components including collagen and fibronectin. Endothelial dysfunction, another common feature of vascular aging, is assessed in vitro by measuring the production of NO or endothelin-1 or by monitoring EC migration and proliferation [79]. Chronic low-grade inflammation, which is a characteristic feature of aging, is assessed in vitro by measuring the production of cytokines and chemokines by ECs and VSMCs [8]. In vitro models of vascular calcification, such as culturing VSMCs with calcifying media, are used to study the cellular and molecular mechanisms underlying this process. Vascular stiffening is evaluated by measuring the contraction force of VSMCs, displacement of the VSMCs relative to a fluorescent microbead, or by assessing migration defects in the cytoskeleton. Tube formation assays, where cells are grown on matrigel to form a tube structure, are used to evaluate EC culture as a model system of vascular aging [80].

Changes in cellular metabolism are another important hallmark of aging. Mitochondrial function is typically assessed using the Seahorse XF Analyzer (Agilent, Santa Clara, CA, USA), which measures cellular respiration and glycolysis in cell culture. Furthermore, gene expression analysis is often combined with the previously mentioned experiments to identify genes that are differentially expressed in aged versus young cells in 2D culture. These genes include markers for senescence, immune infiltration, phenotypical switching of VSMCs, and DNA damage.

### Limitations Cell Culture Models to Study Vascular Aging

Overall, cell culture methods provide insight into cellular processes involved in aspects of vascular aging. However, in vitro culturing of cells can induce chemical and physical alterations, thereby not fully reflecting all processes that are involved in vascular aging [22]. Moreover, the vasculature comprises diverse cell types that engage in constant interactions and communication. Restricting studies to a single isolated cell type may oversimplify the complex nature of vascular aging. To better replicate the intricate dynamics of vascular aging, advanced microfluidic cell culture systems are employed. These systems offer a more comprehensive approach to investigate the complex interactions among different cell types within the vasculature. By integrating multiple cell types and simulating physiological conditions, microfluidic platforms provide a valuable tool for studying the multifaceted aspects of vascular aging.

## 8. Microfluidic Cell Culture Systems and Vascular Aging

The field of microfluidic technology is rapidly advancing, offering increasingly biomimetic in vitro models to complement animal models. Vascular microfluidic devices, which mimic blood flow, are highly adaptable in terms of shape, surface coating, and network complexity. These devices will be a great addition in studying vascular aging where cells with specific aging related mutations can be used to mimic vascular aging. While traditional in vitro culture models are comparatively simplistic, microfluidic models do require a certain level of complexity in their fabrication process. Co-culturing cells of different origins poses a challenge, especially when combined in a 3D culture setting, but is essential to capture the interactions between different cell types, especially in the context of the vascular system. The advantages of microfluidic in vitro systems include their introduction of hemodynamic stimuli (such as shear stress) and interaction with circulating cells, as well as their compatibility with multiple analytical techniques and high throughput data generation and processing.

Microfluidic systems aim to recapitulate the mechanical, electrical, chemical, and structural features of the in vivo microenvironment. Compared to current in vivo or 2D culture conditions, they allow for more precise control of factors such as blood flow, blood pressure, and chemical gradients while providing a reproducible system. Microfluidic cell culture systems can help generate valuable insights by measuring cellular changes with age such as proliferation, migration, or differentiation.

In this review, we will describe four different strategies of microfluidic modelling as depicted in Figure 5, namely, microfabricated channels (a), microfluidic vessel systems (b), self-assembly networks (c), and cell patterning microfluidics (d).

### 8.1. Microfabricated Channels

Microfabricated channels are the most basic form of microfluidic modelling (Figure 5a), yet they still provide a more physiologically relevant environment for cells compared to traditional 2D cell culture models. To create artificial vessels that mimic the specific dimensions and geometries of blood vessels, small channels can be etched or molded into biocompatible substrates like glass or synthetic polymers. Glass, silicon, and polymers are substrates that are often used to make a channel surface to replace natural ECM for cell attachment. These systems are best suited to evaluate the endothelial response to variations in flow (shear stress) and can be coupled to the analysis of endothelium derived factors released in the circulating medium.

An example of such a system is the microvascular network of Borenstein et al., who created circular polystyrene channels with bifurcation in which they seeded ECs to form a monolayer, with the ECs remaining viable for at least 24 h [81].

Another example is the use of glass etching as shown by Rodriguez et al. In one of their approaches, they use a polydimethylsiloxane (PDMS) stamp to form the desired structures on a glass slide [82]. Once the scaffold is complete, ECs are cultured on a specified membrane, in a designed pattern. However, culture on the stiff surfaces of synthetic materials could be detrimental to natural vascular cell behavior and function.

### 8.2. Microfluidic Vessel Models

Recent advances in bio-fabrication techniques allow the engineering of complex vascular networks in vitro that aims to mimic in vivo vasculature as closely as possible [83]. Microfluidic vessel systems (Figure 5b) approximate the structure and function of blood vessels even closer than microfabricated channels. These systems are generated using materials such as (fully synthetic or semi synthetic) hydrogels and are designed to have specific dimensions and geometries that mimic the network structure of blood vessels (Figure 5b) [84]. These models allow for co-cultures of, e.g., ECs with VSMCs, pericytes, and fibroblasts in a 3D extracellular matrix environment. An example of such a model is van Dijk et al.’s PDMS device with a reservoir for a 3D fibrinogen gel with pericytes. This multi-cell type device allows the interaction of ECs with pericytes in full bio-matrix encased 3D vessel structures (neovessels) that can be subjected to continuous, unidirectional flow and perfusion with circulating immune cells [85]. Another example is presented by the research of Polacheck et al. in which they designed and fabricated microfluidic channels with dimensions mimicking small blood vessels. Similar to the van Dijk system, the channel surface provides the ECM environment for EC attachment and expansion to replicate the vascular endothelium. The fabricated microchannels are used to investigate the effects of fluid flow and shear stress on ECs, providing insights into vascular physiology and pathology [86]. Such systems are often produced as small units that can be flowed in serial or parallel configuration and can be scaled up to study many conditions in high throughput fashion.

### 8.3. Microfluidics Combined with Self-Assembly

The self-assembly (Figure 5c) technique differs from the other methods as it does not require a pre-designed structure to direct the growth of cells. For this approach VSMCs are seeded into hydrogel, where conditions are optimized to stimulate spontaneous sprouting and 3D vascular network formation, more closely mimicking the in vivo development of vessels.

An example of this system is the microfluidic device of Wang et al. [87], which uses two microfluidic channels which are coated with laminin and lined with a monolayer of ECs. These channels (representing artery and vein) are connected by a middle tissue chamber filled with an ECM-based hydrogel, in which a new capillary network will form via sprouting of the endothelium in the channels. This system can be perfused with only minimal leakage. Another self-assembly system is described by Vila Cuenca et al. in which multiple cell types are seeded in a fibrin hydrogel microenvironment. In this system, human iPSC-derived EC and VSMCs are used, with cells self-organizing in the fibrin matrix into lumenized and functional vessels. Notably, the created vessel responds to vasoactive stimulation, which can be measured [88].

### 8.4. Cell Patterning

Dependent on the parameters under investigation it can be advantageous to apply cell patterning rather than self-assembly. In the field of tissue engineering, the strategy of cell patterning (often) involves material driven solutions that allow predefining cell organization within an “aggregate” module, with multiple modules constructing the engineered tissue. On a higher design level, cell patterning can also involve the organization of either self-assembled cell units or “patterned” modules into a predesigned configuration to mimic organ structures or to allow more effective self-assembly into higher organization structures (Figure 5d). An example of this strategy is provided by Garvin et al., who used an ultrasound standing wave field to induce patterning of cells in an engineered tissue construct [89]. This technique can determine the speed of formation, density of the cells, and the morphology of the vascular network [90]. Another example of cell patterning is the use of a PDMS stamp to induce cell elongation. These stamps contain strips of fibronectin in a microfluidic channel. Cells are seeded and non-adhesive cells are washed off, resulting in alignment of the cells in one direction [91].

### 8.5. Scientific Outcomes Microfluidic Systems

Various applications have emerged for utilizing vasculature-on-a-chip in in vitro disease studies, particularly focusing on endothelial dysfunction. This vascular condition involves impaired vasomotion control, linked to increased inflammatory responses and vascular leakage. While chip systems mainly investigate inflammatory responses, thrombogenic reactions, flow disturbances, and endothelial permeability in endothelial dysfunction, there is a gap in exploring endothelial-dependent vasomotion dysregulation [89,92,93]. This gap is mainly due to the high stiffness of chip materials, limiting the contractile behavior of VMSCs. Adjusting substrate stiffness could pave the way for future chip designs addressing the defining aspect of this vascular disease. Another area of research in which vascular chips have proven useful is cancer research. Tumor angiogenesis signifies the orchestrated development of new blood vessels crucial for supplying nutrients and oxygen to cancer cells. Integral to tumor progression, it involves the formation of aberrant and intricate blood vessel networks within and surrounding the tumor tissue. These processes have been investigated with the use of various chip systems via tumor and vascular cell co-cultures [94,95,96]. Similarly, cancer metastasis, representing the spread of tumor cells from the primary tumor site to distant organs or tissues, has been studied in complex multi-organ-chip systems incorporating vascular cells to mimic trans-endothelial migration [97,98]. This multi-organ approach may provide a more representative system for drug screening, which has, thus far, used less complex microfluidic systems for visualizing biological and mechanical dynamic processes. These processes include adaptations in cell–cell junction stability, viability, and proliferation rate, as well as vascular cell response to altered shear stress when exploring the efficacy of pharmaceutical compounds [99,100,101].

Future chips should explore the integration of vasomotion regulation, complex perfused vessel building, and co-culture with tissue/tumor cells in multi-organ designs, in combination with an increase in scale and reproducibility. This will allow further implementation in pharmacological studies and application in wider research fields, including (vascular) aging.

### 8.6. Comparison Microfluidic Systems

Microfabricated channels are more straightforward compared to other systems, making them quicker and easier to use. They often use only one cell type, while the other systems allow for a co-culture, closely mimicking the in vivo situation. The other three systems allow the formation of multiple layers of cells in ECM. Microfluidic vessel systems focus on replicating the structure and function of blood vessels or vascular networks, primarily for vascular biology studies and tissue engineering applications. This system works very well for co-culturing larger structures and is, thereby, suitable to study arterioles. The self-assembly microfluidic system is more suitable for investigating the capillary system. Cell patterning microfluidic systems, on the other hand, are designed to precisely pattern and arrange cells, enabling the study of cell behavior, tissue development, and organ-on-a-chip platforms. A disadvantage of this system is that the majority of these only have a limited throughput [102].

## 9. Personalized Medicine

The use of cell culture and microfluidic devices also allows for precision medicine, in which we consider individual differences such as gender, genetics, and lifestyle [103,104]. With precision medicine, a large group of patients can be divided into clusters based on patient characteristics such as clinical phenotype, environment, and molecular analysis, and receive specialized treatment. Examples of how cell experiments and microfluidic devices can be used in future are by identifying new biomarkers, predict clinical outcome, and determine disease susceptibility.

Disease modelling and prevention based on human cells have the advantage that they do not require translation from a mammalian model organism to the human situation, and they can also potentially be personalized for each patient. This personalization of medicine can be facilitated using human induced pluripotent stem cells (hiPSC) derived from healthy individuals or patients. These hiPSCs can be utilized to study disease mechanisms and develop targeted therapies. By using CRISPR Cas9 technology, mutations causing DNA repair deficiencies can be introduced into control hiPSCs, allowing for the investigation of different mutations in an isogenic background. By differentiating these iPSCs into various cell types, such as human ECs, pericytes, and VSMCs, researchers can study the underlying mechanisms of genomic instability in vascular cells. The mutations of the previously mentioned mouse models are good candidate genes to target. Once the necessary cell types are generated, they can be transferred into microfluidic systems to compare specific drug responses within the controlled environment of a high throughput system [105,106]. The use of microfluidics systems can also be advantageous to model human age-related disorders that may manifest differently in humans compared to model organisms. Microfluidic systems are already used in preclinical drug testing. Luna et al. use a microfluidic device to assess blood thrombosis and study anti-thrombotic therapy [107]. They showed blood clotting under flow in tortious arteriolar vessels. Another advantage of using iPSCs in microfluidic devices is the opportunity to test the effect of a mutation in a specific cell type. This information can be used to determine which mouse models to develop. Studying vascular aging in a microfluidic system will be a new and interesting step forward in the use of microfluidic models. However, limitations exist for the use of microfluidic systems. A disadvantage of microfluidic systems compared to in vivo systems is the lack of whole-body integration, such as the lack of hormone circulation, an immune system, and the influence of the central nerve system. Furthermore, microfluidic systems are limited scale; the size of microfluidic vessel-on-a-chip systems is much smaller than actual blood vessels in the human body.

## 10. Conclusions

In summary, vascular aging is linked to genomic instability, as observed in patients with progeria. To better understand age-related vascular changes, researchers have developed mouse models with genome instability that exhibit accelerated vascular aging. These models offer an efficient experimental system to investigate the mechanisms, pathways, and interventions associated with vascular aging. They provide time efficiency, facilitate the discovery of molecular pathways, allow for intervention testing, and offer translational relevance and biomarker discovery, enhancing our understanding of vascular aging and potential therapeutic approaches. However, it is important to note that there are prominent differences between progeroid mice and naturally aged mice or humans. To address this, human vascular cells can be utilized in cell culture studies and microfluidic systems. High complexity microfluidic cell culture systems are powerful tools to study vascular aging as they resemble the structure and function of blood vessels and provide the possibility to work with human cells of the vessel wall. These systems provide precise control over the experimental environment and the cells under investigation. Combining research of microfluidic systems with accelerated aging mouse models offers the opportunity to uncover the underlying mechanisms of vascular aging, bridging the gap between mouse models and human.

## Figures and Tables

**Figure 1 ijms-24-15379-f001:**
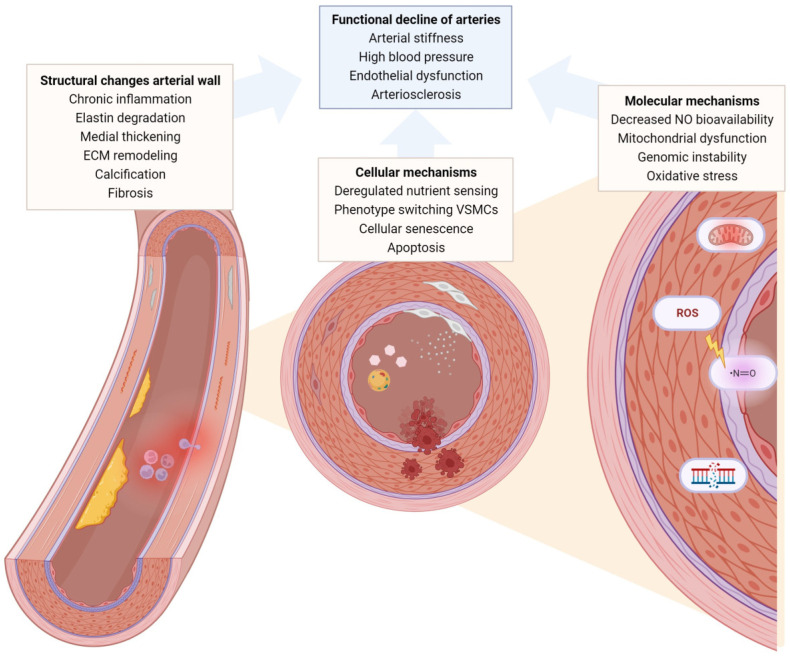
**Structural and functional changes in vascular aging.** Vascular aging is characterized by functional decline of the arteries. Structural changes in the arterial wall together with various cellular and molecular mechanisms contribute to arterial functional decline.

**Figure 2 ijms-24-15379-f002:**
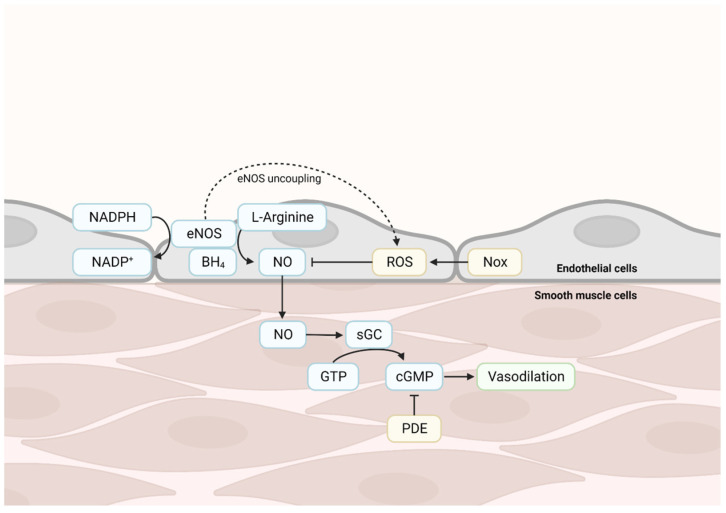
**Nitric oxide (NO)-cyclic guanosine monophosphate (cGMP) signaling and vascular dysfunction**. In blue, important mediators of NO-cGMP signaling; in yellow, factors capable of perturbing NO-cGMP signaling. GTP, guanosine triphosphate. In green, these processes are leading to vasodilation.

**Figure 3 ijms-24-15379-f003:**
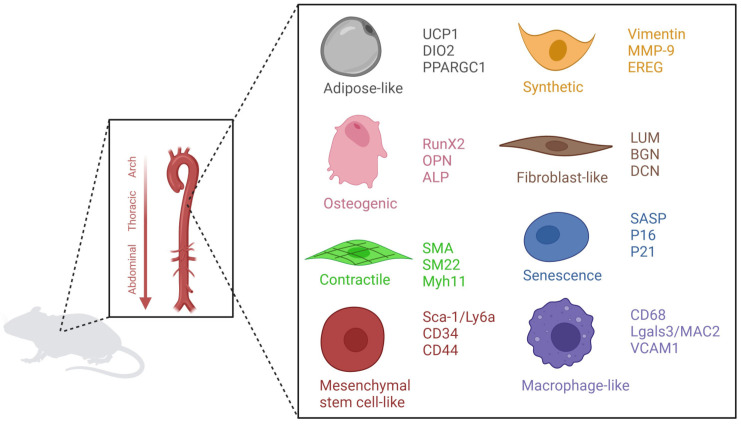
**Vascular smooth muscle cell phenotypes**. These phenotypes are adipose-like, osteogenic, contractile, mesenchymal, stem cell-like synthetic, fibroblast-like, senescent, and macrophage-like VSMCs.

**Figure 4 ijms-24-15379-f004:**
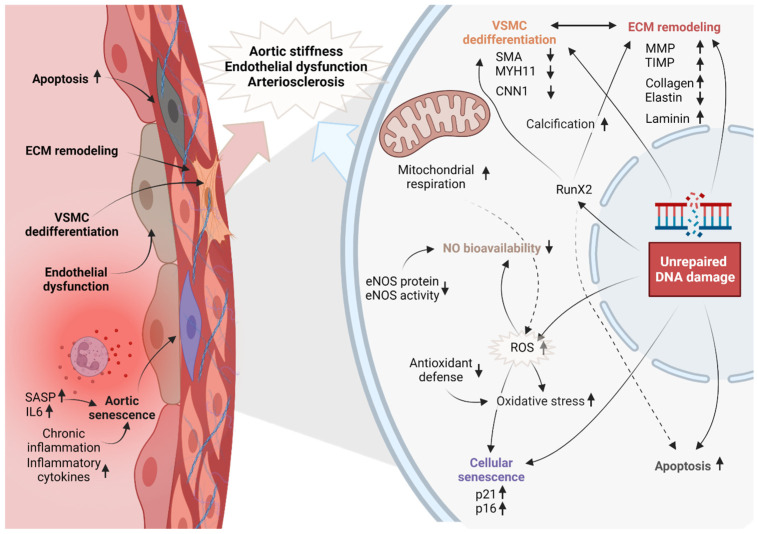
**Parameters of vascular aging and the role of genomic instability.** Accumulation of DNA damage results in structural and functional changes in the vascular cells and the arterial wall.

**Figure 5 ijms-24-15379-f005:**
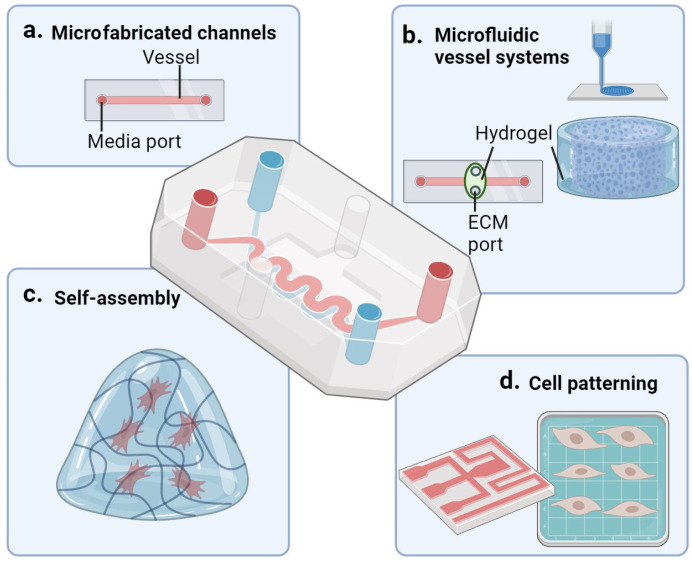
**Schematic representation of microfluidic fabrication methods.** (**a**) Microfabricated fluid-channels. (**b**) Microfabricated blood vessel systems, e.g., bioprinting and sacrificial molds. (**c**) Self-assembled network within hydrogel. (**d**) Patterned microfluidic networks.

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
