# Peer review of "Model Systems to Study the Mechanism of Vascular Aging"

_ijms, 2023, doi:10.3390/ijms242015379_

Round 1

Reviewer 1 Report

The article shows an interesting revision of the most used models to studya accelerated aging process (progreria syndromes in humans). The figures as good to clarify information included in the text. I am recomending the publication ot the article in its present form. 

Author Response

Dear reviewer,

Thank you for reading our manuscript. No comments were suggested to address, we therefore only changed the manuscript according to the other reviewers comments.

Best,

Janette van der Linden

Reviewer 2 Report

Linden et al. submitted a review manuscript focusing on model systems to study the mechanism of vascular aging. Overall, the review looks good as it discusses different study models that may be good for researchers to better understand the systems for using in the studies. However, there are some points that need to be improved. 

·        In addition to nitric oxide, please also include other endothelial mediated pathways (e.g., EDH, PGI2) for vascular regulation, and describe the effects of aging on them. 

·        Please provide appropriate references to each phenotype of smooth muscle cells as mentioned (line 110-112). 

·        Also, update the figure 3 and the legend with appropriate names as in the text (e.g., macrophage-like, mesenchymal stem cell-like, fibroblast-like and adipose-like). 

·        There are several statements throughout the manuscript that need references. Authors are encouraged to go through the manuscript and add more references where necessary. 

·        Please make a list/table of different cell types that are used in vascular aging research. It may be good to write a separate section for limitations of in-vitro cell cultures. 

·        What about age-related disease models? For examples, mCAT mice, Alzheimer’s disease models etc.? Authors might want to write a section about age-related disease models as well. 

·        Along with their design and the ways they work, please also describe the success of using these microfluidic cell culture systems in vascular research. What are the scientific outcomes that utilized these techniques?

Author Response

Dear reviewer,

Thank you for reading our manuscript. We have incorporated most of the suggestions.

Please see below the response of the auteurs, highlighted in yellow. All line numbers refer to the revised manuscript file with tracked changes.

Best,

Janette van der Linden

Reviewer 3 Report

Ther manuscript, model systems to study the mechanism of vascular aging,  by Linden JVD et l  reviewed: (1) Physiological change in the aged vascular system;  (2) Vascular aging in human progeria  syndrome; (3) Vascular aging mouse models; (4) Mechanism of vascular aging in mice; (4) Models to study vascular aging in vitro; (5) Cell culture model  for vascular aging, including microfluidic cell/vessel culture,  cell patterning; (6) Personalized  medicine during aging. This review is interesting. The following comments may be helpful to further understand the structural and functional changes in vascular aging; and the in vivo and in vitro studies of vascular aging and personalize medicine in aging.

1.       It is interesting to add pathological/ structural remodeling in vascular aging before the section of physiological change/remodeling in vascular aging.

2.       It is also interesting to add a section of addressing the molecular and cellular mechanisms underlying vascular aging.

3.       Please point out the advantages and disadvantage of each mouse model for studying arterial aging.

4.       Other vascular aging animal models such as rat, nonhuman primates could be described in brief.

5.       It is important to address how each cell/vessel culture models included here are suitable for the study of vascular aging.

6.       Structural and functional remodeling such as intimal medial thickening, matrix restructuring etc are overviewed in Figure 1.

7.       Please point how NO derived from ECs affects the phenotype of SMCs in Figure 2.

8.       It is interesting to illustrate the molecular inflammatory signaling pathways to each phenotype of smooth muscle cells described in Figure 3.

9.       Please add review how vascular aging contributes to the personalized /precision medicine.

It is OK.

Author Response

(The authors gave the same response as above.)
